

# Consumption of safe drinking water in Pakistan: its dimensions and determinants

**NaeemAkram**
Assistant Chief, Economic Affairs Division, Islamabad
Email: naeem378@yahoo.com
Mobile: +92-333-5343163

**Abstract**

Access to clean and safe drinking water is a basic human right. Poor quality of drinking water is directly associated with various waterborne diseases. The present study has attempted to analyze the household preferences for drinking water sources and the adoption of water purifying methods at home in Pakistan by using the household data of Pakistan Demographic and Health Survey 2017-18. It has been found that people living in rural areas, headed by aged ones and having large family sizes are significantly less likely to use safe drinking water sources and households having media exposure, education, women empowerment in household purchases and belonging to the rich segment of society are more likely to use safe drinking water source. Similarly, households belonging to urban areas, having a higher level of awareness (through education and media), belonging to wealthy families, women enjoying a higher level of empowerment and using piped water are more likely to adopt water-purifying methods at home. However, households using water from tube wells, wells, and boreholes and having higher family sizes are less likely to adopt water purifying methods at home.

**Key Words**:       Drinking Water**,** Education, Filtration, Health

**JEL Classification:**  D31, I26, J31





# 1. Introduction


Access to clean and safe drinking water is a basic human right. However, due to population
growth and limited resources, in developing countries, the utilization of contaminated water is
increasing. Approximately 12% of the world population lacks access to safe drinking water
(World Economic Forum 2019). WHO had estimated that over 2 billion people worldwide do not
have access to drinking water free from contamination at their homes;  among them, 263 million
people have to spend at least 30 minutes to reach water source and 159 million people get
drinking water from rivers, streams or lakes (WHO/UNICEF JMP 2017).
Consequently, millions of people are suffering from chronic diseases like typhoid, diarrhea,
cholera, and parasites because of drinking contaminated water (Curry 2010).  It had estimated
that due to diarrhea, around 1.3 million people die annually; among them 88% are children and
most of these fatal diarrhea cases are associated with poor quality of water and sanitation (IHME,
2015). Usage of safe drinking water leads to reducing the water borne diseases including
diarrhea (Fewtrell et al.2005). It is supported by the fact that during 1870-1930 due to the
provision of piped water in the urban areas of the USA, mortality rates had declined rapidly
(Cutler and Miller, 2005).  However, Brick et al. (2004) and Checkley et al. (2004) were of the
view that health benefits from clean water can only be achieved if there are better sanitation and
hygiene conditions available. Bad hygiene at places of newborn babies along with unsafe water
results in infectious diseases that are the major source of deaths of newborns and 25% of these
deaths can be prevented by providing safe water and sanitation at the place of birth
(IGME,2019).
Pakistan ranks 9[th] in the list of top 10 countries without access to safe drinking water; in
Pakistan, approximately 21 million out of 207 million (total population), do not have access to
safe drinking water (Water Aid, 2018). Similarly, the Pakistan Council of Research in Water
Resources (PCRWR, 2012) concluded that over the years, the quality of water has deteriorated
because of the contamination of chemical pollutants and human waste. It also asserts that in
many areas piped water also polluted due to leakages and its closure to sewerage lines. The poor
quality of water is the main cause of around 60% of infectious waterborne diseases in Pakistan
(WHO, 2008).

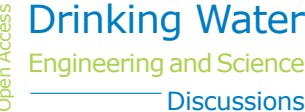

The provision of clean water to the households can be achieved in two ways: by supplying
treated water at the point of gathering or treating water at the point of use. In the first approach,
studies found the significant contamination can occur during the process of transportation and
storage of the water and even storage material and duration affects the water quality (Checkley et
al. 2004, Brick et al. 2004). Brick et al. (2004) and  Fewtrell et al. (2005) are of the view that
treating water at the point of use is the more effective method for the provision of safe drinking
water as compared to supplying treated water at the point of gathering. Even very simple
methods like the use of plain cloth can clean the water to some extent (Colwell et al. 2003).
Mintz (1995) and Quick et al. (1999) concluded that boiling and chemical treatment can
eliminate bacteria but these are relatively costly methods.  Chlorination is considered one of the
cheapest and effective methods for household water treatment (Clasen et al, 2015). However,
various studies concluded that despite having positive impacts very limited households use in-
house water purifying methods (Brown and Clasen, 2012).
In Pakistan, there are numerous sources of drinking water including wells, hand-pumps, piped
water, tube wells, ponds, rivers, bottled water, and fountains, etc. Similarly, different
methodologies like boiling, use of charcoal, filters, etc has been used to treat the water at home.
Consumer behavior regarding the use of safe drinking water is affected by numerous factors. In
this regard income, education, age, household size, level of awareness, number of children and
gender of household head are among the key factors in determining the consumption of safe
drinking water in Pakistan (Sattar and Ahmad, 2007, Rauf et.al 2015, Zulifqar et.al, 2016).
The present study is an attempt to analyze the household preferences for drinking water sources
in Pakistan and adoption of purifying methods at home. Furthermore, the impacts of different
socio-economic factors on household consumption of drinking water and purifying methods will
be analyzed.

## 2.  Literature review

Numerous studies have been conducted to analyze the role of different socio-economic factors
on the consumer choice of drinking water; a brief overview of the selected studies is summarized
as under:



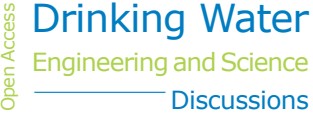

Bruce and Gnedenko (1998) find that income, locality of residence, perception about water
quality significantly affects the use of different water purifying methods. Abrahms, et al. (2000)
finds that water quality (odor, taste), perceived risk of using tap water, age and race are
important factors in the usage of bottled water. Whereas, perceived risk of water-borne disease
and income determine the use of water filters.
Dasgupta (2001) and Mc-Connell and Rosado (2000) found that the level of education positively
and significantly affects the household's consumption of purifying drinking water at home.
Similarly, according to Jyotsna et al (2003) in comparison to media exposure and education
wealth is a stronger factor in determining water purification behavior; furthermore, households
with a higher level of female education are more willing to pay for clean drinking water.
Quick et al. (1999), Mintz et al. (2001) Jalan and Somanathan (2008) and Jalan et al. (2009)
comes to the conclusion that awareness about the health hazarded associated with the use of
unsafe water, cost of treatment, wealth and education have significant impact on purifying
drinking water at home. Fotue Totouomet et al., (2012) and  Daniel et al (2019) found that the
wealth of the household, Education and facing the risk of water-borne disease are the major
factors in determining the adoption of in-house water purifying methods.  Households that are
using piped water are having a higher probability of using purifying methods at home to clean
the drinking water.
In Pakistan, Haq, et al. (2007) are of the view that household locality (urban/rural), education
and quality of available water plays a significant role in determining the demand of improved
water source. Sattar and Ahmad (2007) found that the education of household head and
exposures to media have a significant impact on the choice of different water purifying
methodologies. It was also been found that wealthier people prefer to use expensive technologies
like filters. Furthermore, the education of households has a much stronger effect as compared to
the income level.
Rauf et al (2015) found that family size, distance of the house from the water source and lack of
transportation has a significant and negative impact on the choice of safe drinking water. The
study also found that wealth, and living in urban area has a positive and significant relationship

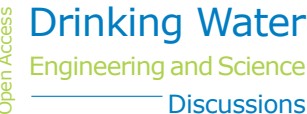

with the choice of safe drinking water. However, the study found that education and gender of
the household head have an insignificant relationship with the choice of safe drinking water.
Zulifqar et.al, (2016) concluded that per capita income, living in urban areas, the awareness level
has a positive impact on the choice of safe drinking water. However, it has been found that the
age of household head and the incidence of water-borne disease to any household member have a
negative relationship with the choice of safe drinking water.

## 3. Methodology


In the present study, the data of Pakistan Demographic and Health Survey (PDHS) 2017-18 has
been used. DHS surveys are conducted in different developing countries with the funding of the
United States Agency for International Development (USAID). In PDHS 2017-18; 15,068
households were selected. In the household survey, we have available information regarding the
source of household drinking water as well as the treatment measures adopted by households to
clean the water.
In survey 17, drinking water sources had been mentioned. To examine the role of different socio-
economic factors in determining the water source, the Multinomial Logit (MNL) model will be
used. The reason is that our dependent variable does not have any ordering and they are multi-
categories. By using MNL, we will examine the preference for different drinking water sources
by using the Filter/ bottled water as the base category. Similarly, Logit Model would be applied
to analyze whether a household applies any measure to clean the water at home or not. In this
regard, a binary variable is created that takes the value of 1 if the household adopts any treatment
method and zero otherwise. The independent variables are distance to the water source,
household wealth, education, exposure to media (a proxy for the level of awareness), household
size, urbanization, etc. Both models have been estimated by using STATA 13.0. A brief
overview of the variables that are used in the analysis is summarized as under:
**i.   Source of Drinking water**
In the survey, there are 17 different water sources. However, depending upon the nature of these
sources we had grouped them into 6 different water sources. These are 1) Filtration plant/Bottled



water, 2) Piped Water, 3)Tube well / borehole/ protected well, 4)Unprotected well/springs
5)River/Dam/Lakes/ Ponds/Canals/ Streams, 6)Tanker/ Truck/ Carats with small tank.

### ii.   Adoption of any purifying method to clean the water

We had created a binary variable to represent purifying methods used by the households. It takes the value of 1 if the household adopts any type of purifying method at home and 0 if the household does not adopt any purifying method.

### iii.   Age of household head

The age of household head can be an important factor in determining the water source as well as the purifying method. It is expected that households headed by more aged ones are less likely to use safe drinking water and adopt modern purifying methods. It is categorized as15-25, 25-39, 40-59 and 60 or more years of age.

### iv.   Level of education of household head

Numerous studies had recognized that education plays a pivotal role in choosing a safe drinking water source. In the dataset, education is divided into four categories no education, primary, secondary and higher education. We expect that education will positively affect the choice of safe drinking water sources and the use of purifying methods.

### v.   Household Size

It is expected that household size will hurt the choice of safe drinking water as well as the usage of any water purifying method. This variable is categorized as the family size of 1-5, 6-10, 11-15 and 16or more members.

### vi.   Wealth of household

The wealth index had been used to describe the wealth of the household. The wealth index is calculated by using the principal component analysis of around 40 different asset variables including the housing facilities, consumer and other material. The wealth index can take value from 1-5 where 1 indicates the poorest and 5 as the richest household.

### vii.   Exposure to media



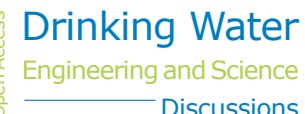

We constructed a binary variable named exposure of media (reading newspaper, watching TV or
listening to the radio). It takes the value of 1 if a household either reads the newspaper, watches
TV or listens to the radio, indicating that the household has exposure to media. Value of 0
represents no media exposure, the variable takes a value of 0 if he does not use any form of
media.
**viii.    Women Empowerment**
There are several aspects of women empowerment. These include control over resources,
involvement in household decision-making, and economic contribution in the household,
freedom of movement, sense of self-worth, appreciation in the household, time use, knowledge,
division in household work, etc. Keeping in view the nature of the present study, we had used
only her autonomy in household purchases as an indicator of empowerment. In the dataset the
question had five responses 1) respondent alone 2) respondent and husband/partner 3)
husband/partner alone 4) family elders and 5) others. To make binary variables in the study, the
first two responses are assigned the value of 1 describing that woman has autonomy and 0 for the
rest of three options indicating that she had no autonomy.
**ix.    Distance to the water source**
In the original data set, there is no direct variable available that measures the distance to the
water source. However, there is a variable that gives the details of the time (round trip) to get to
the water source. It is used because if the water source is far away then it will take more time as
compared to the availability of water nearby. The variable is having three options, 1) water is
available at home 2) It takes up to 15 minutes to reach water source 3) It takes more than  15
minutes to reach a water source.
**x.    Locality**
Rural and Urban areas are two bifurcations of the locality. In this regard, a binary variable has
been constructed assigning a value of 1 for rural households and 0 for urban households.


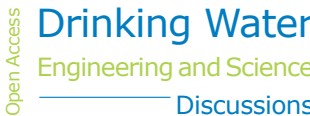

# 4. Results and Discussions

Before conducting econometric analysis, descriptive statistics of variables are presented in Table
1. It suggests that 48% of the surveyed households were living in urban areas while around 52%
of the sampled households were living in rural areas.
**Table 1 Descriptive statistics of explanatory variables**

| Variable | Mean | Proportion | Std. Dev. | Minimum | Maximum |
|---|---|---|---|---|---|
| **Locality** | 0.48 | | 0.50 | 0 | 1 |
| Urban | **** | 48.1% | **** | | **** |
| Rural | **** | 51.9% | **** | | **** |
| **Water Source** | 2.81 | | 0.99 | 1 | 6 |
| Filtration plant/Bottled water | **** | 5.5% | **** | **** | **** |
| Piped Water | **** | 32.0% | **** | **** | **** |
| Tube well / bore hole/ protected well | **** | 46.7% | **** | **** | **** |
| Unprotected well/springs | **** | 10.5% | **** | **** | **** |
| River/Dam/Lakes/ Ponds/Cannels/ Streams, | **** | 2.3% | **** | **** | **** |
| Tanker/ Truck/ Carats with small tank. | **** | 3.0% | **** | **** | **** |
| **Adoption of any purifying method to clean the water Locality** | 0.10 | **** | 0.30 | 0 | 1 |
| No | **** | 89.8% | **** | | **** |
| Yes | **** | 10.2% | **** | | **** |
| **Distance to Water Source** | 0.37 | | 0.70 | 0 | 1 |
| At home | **** | 76.2% | **** | | **** |
| Up to 15 minutes | **** | 10.8% | **** | | **** |
| Above 15 minutes | **** | 13.0% | **** | | **** |
| **Age of Household** | 47.78 | | 14.02 | 15 | 95 |



| Head | | | | | |
|---|---|---|---|---|---|
| 15-25 | **** | 2.4% | **** | | **** |
| 25-39 | **** | 28.5% | **** | | **** |
| 40-59 | **** | 46.3% | **** | | **** |
| 60+ | **** | 22.8% | **** | | **** |
| **Household Size** | 8.43 | | 4.61 | 1 | 44 |
| 1-5 | **** | 26.4% | **** | | **** |
| 6-10 | **** | 50.0% | **** | | **** |
| 11-15 | **** | 16.5% | **** | | **** |
| 16+ | **** | 7.1% | **** | | **** |
| **Education** | 0.99 | | 1.14 | 0 (No Education) | 3 (High) |
| No Education | **** | 50.6% | **** | | **** |
| Primary Education | **** | 14.0% | **** | | **** |
| Secondary Education | **** | 20.8% | **** | | **** |
| Higher Education | **** | 14.6% | **** | | **** |
| **Wealth** | 2.79 | | 1.43 | 1(Bottom 20%) | 5 (Top 20%) |
| Poorest | **** | 25.3% | **** | | **** |
| Poorer | **** | 21.4% | **** | | **** |
| Middle | **** | 19.0% | **** | | **** |
| Richer | **** | 17.1% | **** | | **** |
| Richest | **** | 17.2% | **** | | **** |
| **Media Exposure** | 0.64 | | 0.48 | 0 | 1 |
| No | **** | 35.7% | **** | | **** |
| Yes | **** | 64.3% | **** | | **** |
| **Women Empowerment in Household purchases** | 0.40 | | 0.49 | 0 | 1 |
| No | **** | 60.1% | **** | | **** |
| Yes | **** | 39.9% | **** | | **** |


The majority of the households were drinking water from Tube wells/boreholes/protected wells (47%), followed by piped water (32%), unprotected wells (11%) and water from filtration plant/bottled water (6%) and other sources (4%). Similarly, 90% of households are not using any method to purify the drinking water at home. The majority of household i.e. 76% are getting drinking water at home, 11% of the household have to travel for less than fifteen minutes to

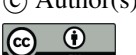



reach water source and 13% of households are getting water from sources where they have to
travel for fifteen minutes or more (round trip). The minimum age of the household head emerged
as 15 years while the maximum age was 95 years and average age of the household head is 48
years.  It is also pertinent to mention that majority of household heads belong to the age bracket
of 40-59 years.  The average family size is eight persons; however, the maximum family size of
the surveyed households was 44 persons and the minimum family size is only one family
member. 50% of the households are having a family size of 6-10 persons. The table also
indicates that 51% of surveyed households were uneducated and only 35% of the household are
having a secondary level or higher education. In terms of wealth, 47% of the households were
poor 19% are among middle and 34% were classified as rich. The table also revels that 64 % of
the surveyed households are having exposure to the media. Similarly, about 40% of the
household's women have empowerment in household purchases.
The study is focused on the determinants of household drinking water source for estimation
Multinomial Logit (MNL) model has been applied. In the MNL model, we had used the water
from filtration plant/ bottled water as the base category. The results are summarized in Table 2
below.
**Table 2 Estimation results of Multinomial Logit (MNL) model of determinants of drinking**
**water source  (relative risk ratios)**

| Variables | Water Sources | | | | | |
|---|---|---|---|---|---|---|
| | Filtration plant/Bottled Water | Piped Water | Tube well/borehole/protected well | Unprotected well/springs | River/Dam/Lakes/Ponds/Canals/Streams | Tanker/Truck/Carats |
| Locality (living in rural areas) | 1 | 1.0094* | 1.1269* | 1.0584* | 0.6082* | 0.0134 |
| Age of Household Head | 1 | 1.2826* | 1.1197* | 1.4915* | 1.0676* | 1.1768** |
| Household Size | 1 | 1.5281* | 1.5405* | 1.3387* | 1.8129* | 1.9999* |
| Media Exposure | 1 | 0.9893* | 1.0989 | 0.7319* | 0.8713 | 0.6348* |



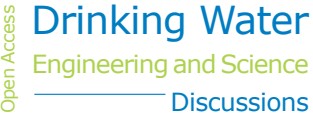

| | | | | | | |
|---|---|---|---|---|---|---|
| Education | 1 | 0.8325* | 0.7136* | 0.6479* | 0.3625* | 0.8397* |
| Women Empowerment in Household purchases | 1 | 0.6489* | 0.7705* | 0.6130* | 0.5478* | 0.3766* |
| Wealth | 1 | 0.4325* | 0.4625* | 0.2505* | 0.3936* | 0.2192* |
| **Constant** | 1 | 110.0963* | 283.4138* | 200.7871* | 10.0194* | 112.5794* |
| **LR Chi-Square** | | 3651.62 | | | | |
| **P-value of Chi-Square** | | 0.0000 | | | | |
| **Pseudo R Square** | | 0.1021 | | | | |

*p < 0.05;   **p < 0.10

The results suggest that urbanization is having a significant impact on the choice of drinking water in four out of five alternatives. The results suggest that people living in rural areas are more likely to use water from protected wells and Tube wells as compared to the water from filtration plant/bottled water for drinking, as the relative risk ratio is 1.13 significantly highest among all the alternatives, (possible reason seems to be the cost and availability of services).  Furthermore, results are also suggestive of the fact that household living in rural areas are less likely to use drinking water from dams/rivers/streams (relative risk ratio less than 1) but they would prefer piped water and also unprotected well/springs (relative risk ratio greater than 1).

The results indicate that age of household head is having a significant impact on the source of drinking water in all the five alternatives.  The results suggest that households headed by aged ones are more likely to consume water from wells, tube wells, piped water, rivers, streams, rivers, dams, tankers, trucks, etc (as relative risk ratios are significantly greater than 1). It reflects that aged people in Pakistan are least health-conscious and they prefer to use traditional water sources instead of water from filtration plants.

Household size is having a very strong impact, as the results are significant in all the five alternatives.  The households having larger family size prefers to use alternatives as compared to the water from filtration plants as in all the alternatives relative risk ratio is significantly greater than 1. This can be due to the larger family size more water is required so families prefer to use water from those sources where they can get more water easily.



It has been found that education (significant in all of the five choices) and exposure to media
(significant in three out of the five choices) have a crucial role in consumption of safe
drinking water. It has been further confirmed that household that is having access to media
and education are less likely to use the water from piped water, wells, tube wells, rivers,
streams, rivers, dams, tankers, trucks, etc (as relative risk ratios are significantly less than 1)
rather they would prefer to use the water from filtration plants. It is because people have
information about the health hazards of unsafe water therefore they would prefer to use safe
drinking water sources.
The wealth of the household emerges another significant factor in the drinking of clean
water. It has been found that wealthier household prefers to use water from filtration plants/
bottled water and they are less likely to use drinking water from piped water, wells, tube
wells, rivers, streams, rivers, dams, tankers, trucks, etc. The reason is quite straight forward
wealthier households can afford the better sources of drinking water. Furthermore, rich
people are more health-conscious and willing to spend more money on an improved water
source.
It has also been found that households with greater women autonomy in making household
purchases prefer to use water from filtration plants/ bottled water and they are less likely to
use drinking water from piped water, wells, tube wells, rivers, streams, rivers, dams, tankers,
etc. It suggests that women are more health-conscious and if they are involved in household
spending decision-making then there are more chances that they would make some cuts in
the budget allocated for makeup and associated luxuries and prefer to spend more money on
an improved water source.
In the next step, the household's use of the in-house water purifying method is analyzed. This
model is tested by using the logit model.  The results are summarized in Table 3.
**Table 3 Estimation results of logit model of the in-house water treatment to treat water**
**(odd ratios)**

| Variables | Odd Ratios | P values |
|---|---|---|
| **Locality** | | |
| Urban | 1 | |



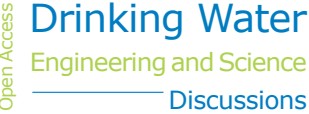

| | | |
|---|---|---|
| Rural | 0.8901 | 0.0569** |
| **Age of Household Head** | | |
| 15-25 | 1 | |
| 25-39 | 0.8677 | 0.459 |
| 40-59 | 0.8805 | 0.505 |
| 60+ | 0.8846 | 0.536 |
| **Household Size** | | |
| 1-5 | 1 | |
| 6-10 | 0.9519* | 0.047 |
| 11-15 | 0.8922** | 0.098 |
| 16+ | 0.8672* | 0.000 |
| **Education** | | |
| No Education | 1 | |
| Primary Education | 1.0702 | 0.447 |
| Secondary Education | 1.1308* | 0.041 |
| Higher Education | 1.8081* | 0.000 |
| **Wealth** | | |
| Poorest | 1 | |
| Poorer | 0.9991 | 0.992 |
| Middle | 0.9005 | 0.266 |
| Richer | 1.0675* | 0.063 |
| Richest | 1.0844* | 0.032 |
| **Media Exposure** | | |
| No | 1 | |
| Yes | 1.1904* | 0.017 |
| **Distance to Water Source** | | |
| At home | 1 | |
| Up to 15 minutes | 1.1270 | 0.253 |
| Above 15 minutes | 0.9610 | 0.722 |
| **Women Empowerment in Household purchases** | | |
| No | 1 | |
| Yes | 1.2291* | 0.001 |
| **Water Source** | | |
| Filtration plant/Bottled water | 1 | |
| Piped Water | 1.0991* | 0.000 |
| Tube well / bore hole/ protected well | 0.5752* | 0.000 |
| Unprotected well/springs | 0.9641* | 0.000 |
| River/Dam/Lakes/ Ponds/Cannels/ Streams, | 0.9984 | 0.994 |
| Tanker/ Truck/ Carats with small tank. | 0.5640* | 0.017 |



| Constant | 0.1608 | 0.000 |
|---|---|---|
| LR Chi-Square (36) | 118.72 | |
| P-value of Chi-Square | 0.000 | |
| Pseudo R Square | 0.0136 | |

$*p < 0.05$;   $**p < 0.10$
The results from table 3 indicate that locality of the household plays a significant role in adoption
of in-house water purifying treatment and people who live in urban areas are more likely to use
the water purifying method (odd ratio for rural households are significantly below 1). Hence,
people living in urban areas would prefer to use water filters and adopt other water purifying
methods at home.
It has also been found that the family size hurts the selection of water purifying methods as odd
ratios are less than 1. Due to the large family size, more water is required so it is not very
difficult for the large families to use water purifying methods rather they prefer to use water
without any treatment. It reveals the fact that due to larger family quality as well as quantity of
essential services negatively affected.
Both the education and exposure to the media (the indicators for the level of awareness) are
having significant impacts on the use of water purifying methods as odd ratios are greater than 1.
It has been further found that only secondary and higher education results in increasing the odds
of adoption of water purifying methods at home. The education up to the primary level does not
have a significant impact on the adoption of water purifying methods.
It has also been found that the wealth of households has a significant impact on the adoption of
the water purifying method. There are significantly higher odds of the wealthier household to
adopt water-purifying methods to clean the drinking water in comparison to a poor or middle-
income household. The women's empowerment is also had a significant impact on adoption of
water purifying method. Household wherein women are empowered in making household
purchases are more likely to use water-purifying methods at home.
The drinking water source is also emerged as an important and significant factor in the adoption
of water purifying methods at home. The results reveal that households using piped water are
more likely to adopt a water-purifying method at home. However, households using water from



tube well, boreholes, protected well, unprotected wells, springs, tankers, truck/ carats with a
small tank are significantly less likely to adopt water purifying methods at home.
However, study finds that age of households and distance to water sources do not have any
significant impact on the use of water purifying methods.

## 5. Conclusions and Policy Recommendations

In developing countries, poor quality of drinking water has been recognized as a major health
issue because many fatal diseases especially diarrhea and hepatitis are linked with the quality of
water. In this regard, IHME (2015) had estimated that due to diarrhea around 1.3 million people
die annually; among them 88% are the children. The study also estimated that these fatal diarrhea
cases are mostly associated with poor quality of water and sanitation. Keeping in view the
importance of safe drinking water for human health and economic development present study is
conducted.  The results of the study provide comprehensive insight for policymakers to tackle
obstacles in the consumption of safe drinking water in Pakistan and it will help them to develop
better initiatives that would increase the availability/usage of better quality drinking water in
Pakistan.
It has been found that locality of household, family size, age of household head, wealth of
household, level of awareness (education and exposure to media), and women empowerment are
significant factors in determining the household consumption of drinking water sources. People
living in rural areas, headed by aged ones, having large family sizes are significantly less likely
to use safe drinking water sources. However, households having media exposure, education,
women empowerment in household purchases and belonging to the rich segment of society are
more likely to use a safe drinking water source.
Similarly, locality of household, family size, education, exposure to the media, women
empowerment, source of drinking water and wealth of household are significant factors in
determining the household adoption of the water purifying method. It reveals that households
belonging to urban areas, having a higher level of awareness (through education and media),
belonging to wealthy families, wherein women enjoy a higher level of empowerment and
households using piped water are more likely to adopt water-purifying methods at home.
However, households using water from tube well, boreholes, protected well, unprotected wells,



springs, tankers, truck/ carats with a small tank and having higher family size are less likely to
adopt water purifying methods at home. However, the age of household head and distance to
water sources do not have a significant impact on the adoption of the water purifying method.
The findings of study suggest that the government along with civil society must regularly launch
awareness campaigns about different methods of safe drinking water. Similarly better drinking
water facilities must be provided in rural areas so that differences in urban and rural areas in
terms of safe drinking water may be eliminated. Furthermore, as it has been found that women
empowerment in household decision-making is another key factor therefore efforts would be
made to empower the women in Pakistan.

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
