# Peer review of "Consumption of safe drinking water in Pakistan: its dimensions"

_Drinking Water Engineering and Science, 2020_

## Referee Comment (RC1) · Anonymous Referee #1 · 26 Mar 2020

**Review on: Consumption of safe drinking water in Pakistan: its dimensions and determinants**

**Summary**

The objective is interesting, but author needs to re-write and re-structure everything if author want to publish this draft. This current format is not suitable for a standard scientific journal. The statistical analysis are also doubtful. Please see my complete comment below.

**Major comment**

Please make the introduction and literature concise. Now you have 3+ pages of it. Please make it maximum 2 pages. That's possible. Delete unnecessary information. Make it concise.

I think the citation and reference's style are not well reported. Please edit it following the journal's standard. Check the example in the website: https://www.drinking-water-engineering-and-science.net/for_authors/manuscript_preparation.html

The conclusion is not strong enough! And the conclusion chapter is not really conclusion, but there are many repetition from the discussion chapter.

Please see other comments for specific chapter

**Minor comment**

Line 4 -> I think it is not necessary to write your job level/position there. The information is also not complete. There is no country name and the name of organization.

**Abstract**

**Important!** : I think that author needs to re-write this abstract. Please consider my comments for other chapters when re-writing the abstract.

Line 12 -> the adoption of water purifying methods at home -> change it to: the adoption of household water treatment (HWT).

-> edit "-18" to "-2018"

-> It has been found that -> this study found that …

-> having a large …

-> safe -> which water sources are safe?

– 18 -> this sentence can be divided into two sentences

-> adopt water purifying methods at home -> change it to: household water treatment (HWT).

**Introduction**

Line 9 –> this is similar to the 1st sentence of the introduction. So re-write this sentence.

Line 34 – 36 -> edit: However, the utilization of contaminated water is increasing in developing countries due to population growth and limited water resources.

Line 36 - 40 -> I suggest to change both citations to:

https://www.who.int/water_sanitation_health/publications/jmp-2019-full-report.pdf

Change the term from safe drinking water to safely managed drinking water and update the value according to the updated report.

Line 41 – 42 -> edit to: consuming unsafe water lead to chronic diseases, such as typhoid, diarrhea, cholera, and parasites (Curry, 2010).

Line 48 – 50 -> please re-write the sentence. I don't fully understand. Also, this phrase seems not good for scientific paper: were of the view that health benefits.

Line 50 -> Bad hygiene at places of newborn babies -> this is not a good scientific terms/words. Please edit it.

Line 50 – 53 ->

Line 53 -> change IGME to doi: 10.1016/j.ijheh.2019.05.004 . Don't use grey literature.

Line 54-56 -> long sentence, divide into 2 sentences. Also put the year of the data. For example: in 2018, there were about 21 million…

Line 56- 58 -> edit to: The Pakistan Council of Research in Water Resources concluded that the quality of water has deteriorated over the years because of the contamination of chemical pollutants and human waste (PCRWR, 2012).
Line 59 -> add "are" between water and also. Also add citation.

Line 63 -> change "point of gathering" to "point of collection". Check throughout the documents.

Line 64 -> change "contamination" to "re-contamination"

Line 66 -> change "are of the view" to "argued"

– 69 -> Delete this sentence: even very …  -> you don't need this sentence.

– 71 -> I suggest to simplify the 2 sentences into: Example of treating water at the point of use are boiling (mintz), chemical treatment (quick), and chlorination (clasen). -> I think you don't need the information of cheaper, etc., because that is not your focus.

– 74 -> re-write this sentence. I think you need comma after "impacts"

– 74 -> change "in-house purifying methods" to simply "household water treatment (HWT)". And then you can use simply use "HWT" for the rest of the draft, so you can reduce the no of words.

– 77 -> this is not necessary. I think everyone knows that in the context of developing countries, there are many types of source. But if you want to keep it, make it concise and you need citation for this.

Line 78 -> what do you mean by the use of safe drinking water? Is that HWT or safe water source? Not clear.

-> you can simply write like this: the use of HWT is influenced by some factors, such as education, income, level of awareness, etc. (sattar,…). -> you can reduce your unnecessary words!

-> edit to : this study attempts to analyse the …

- 85 -> I think those two sentences are similar. Try to combine them and limit your words. Then you can combine this paragraph with the previous paragraph.

**Literature**

Important for this section (!) : please make the section concise. You don't need to write all factors. Just pick factors which are very relevant to the factors that you use.

Line 86 -> edit "literature review" to "factors related to the use of household water treatment"

and 92 -> change "find" to "found" -> the study is in the past

-> add "and" before "perception"

-> affects -> change to affect (plural)

Change water purification and other related terms to HWT. Change purifying water to treated water. Check all the draft.

-> change to "Abraham"

-> comma before "and"

– 94 -> is this sentence coming from "Abraham, 2000"? if yes, why don't you combine them with the previous sentence

-> you can delete "and significantly"

– 99 -> you can simplify these two sentences: moreover, Jyotsna et al. (2003) found that housholds with .. are more willing to pay for ….. -> you can delete the comparison to media …

-> comes to the conclusion -> change to "concluded"

– 103 -> purifying … -> change to HWT

– 107 -> you can add this paper as an extra citation: https://www.nature.com/articles/s41545-018-0012-z

– 107 -> are having a higher probability … -> change to: are more likely to treat water at home compared to ..

-> are of the view that -> I never see this words in scientific paper. Why don't you just say "find", "argue", "imply", "suggest" ? please check all the draft.

-> locality (urban/rural) -> change to "location". Change all the draft!!

-> comma before "and"

-> plays -> play (plural)

112-114 -> I think you can delete these sentences.

-> delete lack of transportation -> you don't use this variable in your analysis right? You can delete it. This is one of the ways to make your review concise. Delete unnecessary info.

– 119 -> delete this. don't need to discuss the insignificant variable here. Why you discuss insignificant variable only in this paragraph and not in the previous paragraph? Be consistent! If you don't discuss it before, you don't need to discuss it here.

-> delete "per capita"

-> awareness level of …

-> choice of safe drinking water -? What do you mean? Water source of types of HWT?

**Summary of chapter 2** : authors can make this section more concise. Don't need to mention all significant variables that are found in those studies, including negative or positive correlation. You can discuss that when you relate your findings and their findings. But don't need to be detail in this chapter. See for example the paper from Daniel et al. (2019) (one of the papers that you cite) how they wrote all the factors very briefly in section 2.2 only in 1 paragraph.

**Methodology**

-> remove "in the present study"

– 127 -> remove this sentence: DHS surveys …

– 130 -> change to: the data on source of .. as well as … to clean the water were used.

-> remove the 1st sentence. Repetition of the previos sentence.

-> "will be used" change to "was used".

– 134 -> change the sentence: that was because the dependent variables are multi-categories -> the dependent variable is one or more than one? If one, use singular. My example is for plural (more than 1 dependent variable).

-> "we will examine" change to "we examined"

-> would be applied -> change to: was applied

-> is change to was

-> "zero otherwise" -> this is not really a good scientific words. change to "zero for not adopting any HWT"

-> "etc." -> I suggest to list all of them

-> is summarized as under -> change: is summarized in the previous section.

**Explanation on variables**:

**Important! :**

1. make clear distinction between dependent and independent variable. For example, you can make different sub-title for them.
2. if you write your assumption or hypothesis in several variables (age of household head and level of education), you need to make it in all variables, e.g., you don't make it for wealth of household.
3. Be careful of using present and past tense! Please check it.
4. i -> change the numbering to 3.1, 3.2, …

– 144 -> I would say that your classification is not scientifically acceptable. For example, why don't you use the old definition of JMP, for example: improved, unimproved, surface water? But I don't want you to re-do all your analysis. So, suggest to change the name for these 6 classifications. For example, no 1: just call it "bottled water", no 3: well, no 4: unprotected well, no 5: surface water, no 6: bough water from commercial entities.

– 154 -> I suggest to delete these 2 sentences and just focus on the categorization.

-> is categorized -> change to "was categorized"

-> delete the 1st sentence. This is repetition of the chapter literature review!

-> we expect -> change to: it was hypothesized that .. -> **change all words** with "expected" to "hypothesized"! -> the word "expected" sounds not a formal scientific words.

-> will hurt -> what do you mean?

-> space between "16" and "or"

- 169 -> I suspect that you did not do the PCA yourself but you just use the output wealth index categorization of the DHS? If this is correct, you need to mention it. also, check my previous comment that you need to be consistent to write also your hypothesis in this variable.

– 175 -> I don't understand this sentence. Please edit it

– 180 -> you need reference for this

-> comma between "datasets" and "the"

-> had change to "has"

– 190 -> these three sentences are unnecessary. Why don't you just change it to: to measure the relative distance to the water source, we utilized the information of walking distance (round-trip) to get to the water source.

-> change "locality" to "location". **Please check the whole draft!**

-> Rural and Urban areas are two bifurcations of the locality -> change to:

**Results and discussions**

**Important! :**

1. the discussion is not deep enough. Author only describe the findings one-by-one for each predictor variable and don't relate the findings to bigger context or other studies. Even there is no discussion which predictor is the most important one.
2. The use of present and past tense are wrong in many cases.
3. Sometimes author write unnecessary words result in long sentence. Please consider to write it more concise.
4. It seems that author treat predictor variable as continuous in multinominal logit but then categorical in logit and this means that the analysis is wrong. Because if the predictors are categorical, table 2 should look like table 3 (all levels in the predictor variables have their own results or they are dummy variables). Please correct me if I am wrong. if I am correct but please re-do the analysis or update the table if I am correct.
5. The statistical analysis looks doubtful. Usually researcher use p value < 0.05, but author also consider p value < 0.1 as significant. Please give your reason for this in the methods section.

-> remove "econometric" because you never mention about it in the methods. You can start the sentence from "Descriptive statistics …

-> change "suggest" to "shows"

Table 1 -> what is the meaning of stars? Also, why the table's color is in black and green? Please edit. You also don't need min and max. re-arrange the table so the sequence of colomn is: variable – proportion – mean – SD

Table 1 -> if you re-code the variable in water source as I suggest above, you can use them here. Also, why water source has mean and SD?

Table 1 -> what is: adoption of any purifying method to clean the water locality? You can simply say this "Applying household water treatment". Also, this variable is categorical, does not have mean and SD! Similar to water source

Table 1 -> education is categorical variable and has no mean and SD! Media exposure and woman eimpowerment as well! I think wealth also because, of course, the mean is around 2.5.

**Table 1 -> in general, this table is poor organized! Please improve it!**

– 220 -> this paragraph is redundant because you also have the information in table 1, including also the sentence in line 200 – 201. You can describe it a bit but not necessary to descrive all variables. Just pick 2-3 important variables, maybe water source and HWT because those two are the output variable.

– 222 -> this sentence is confusing. I think this is two sentences?

Table 2 -> I don't understand why you write "relative risk ratios" in the brackets. I think better to write it in separate sentence: "the value in table is the odds ratios, i.e., Exp(B)."

Table 2 -> change "locality" to "location". "living in rural areas" to "rural areas"

Table 2 -> please use the re-code that I mention above for the water source category

Table 2 -> please use 2 or 3 digit behind the comma. Maybe 2 digit is enough.

Table 2 -> which pseudo R square do you use? I suggest to use Nagelkerke $R^2$. Also, it is not common to show "LR chi$^2$" and "p value chi$^2$". Suggest to delete them.

– 229 -> urbanization -> which variable show urbanization (process)? Urban area does not necessarily mean urbanization (process). The interpretation is different compared to saying: household's location influenced the choice of drinking water. Please edit them. Or maybe just delete them and start from the 2nd sentence.

-> change "are" to "were"

– 233 -> edit sentence: … tube wells compared to other sources (Exp(B) = 1.13 is the highest compared to others). -> stop here and delete the sentences in the brackets.

-> "results are also suggestive of the fact that" change to : "results suggested that … "

-> change "indicate" to "indicated"

-> suggest to "suggested"

– 240 -> wells, tube wells, …. -> this is a long list. Say more specific things. It seems that you just write all the sources. Why don't just say the most important source (highest odd ratio)?

– 242 -> this is a weak "reflection" unless you give a reference, especially this one: "least health-conscious". How do you know that? Also, you can not prove that the filtration plants are safer than wells, etc.

– 246 -> I don't understand this sentece. What do you mean by "prefer to use alternatives…"? or do you mean compared to bottled water? If yes, make it clear.

-> This can be due to the larger family size more water is required -> this is confusing sentence!

– 250 -> this is "empty" sentence because it seems that all variables are crucial. Re-write or delete.

-> it has been found -> I never see any scientific papers wrote like this. You can write something like these: we found.. , XXX was found positively influencing .., the results shows that … -> **Please check the discussion because I found some cases like this!**

– 251 -> again, long list of sources. Can you make it

– 255 -> if you write "it **is** because", it seems that your reason is 100% valid. but of course not because you don't have any analysis about that and you only assume. So you can write: it may be because… . Also, add citation to support your argument.

– 257 -> another "empty" sentence

– 259 -> again, unnecessary and long list of water source! Why don't you just write: … prefer to use bottled water than other water sources. -> similar comment to line 265

-> the reason is quite straight forward -> come on! Is this a scientific words?

-260 -> again, if you use "is" and "are" for reasoning, it seems that you are 100% sure about that. But you don't have data/analysis. Re-write this sentence! Also, why don't you cite other findings which can support you assumption / argument? How can you be totally sure about your assumption without any supporting data/literature?

– 269 -> "then there are more chances that they would make some cuts in the budget allocated for makeup and associated luxuries … " -> what is this?! You got this information from newspaper?

- 271 -> "is" to "was"

Table 3 -> make the number to 2 digits behind the comma

Table 3 -> the number sometimes are in the middle, sometimes are in left or even right. Please improve and be consistent!

Table 3 -> which pseudo R square do you use? I suggest to use Nagelkerke $R^2$. Also, it is not common to show "LR chi$^2$" and "p value chi$^2$". Suggest to delete them.

Table 3 -> please use the re-code that I mention above for the water source category

Table 3 -> why p value 0.000 is not

Table 3 -> check again all the comments for table 2 and improve this table.

-> locality -> location

-> **in-house water purifying treatment -> change it to HWT. Please check all the draft!**

-> why do you mention water filter here and not other HWT methods? Why don't just say: "… prefer to treat water at home."?

– 281 -> "it has also been found", "hurts", water purifying methods" -> edit as I mentioned previously.

– 283 -> I don't understand this sentence. Why don't you separate them? Also it seems counter-intiutive because I think that the larger the family, the more difficult them to treat water, i.e., don't have time/money/resources to treat larger amount of water. Your reason is not really strong.

– 284 -> very confusing sentence! Even I cannot understand the translation in my language when I use google translate. Please edit!

– 286 -> similar comment to line 248 – 250 previously.

-> it has been further -> ….

– 289 -> So what is the conclusion of these two sentences? Why don't you just write that: the higher the education, the more likely households to treat water" ? it is more simple than two "empty" sentences.

- 300 -> so what can you say from these two sentences? You can write something like: the results indicate that people might not trust the water quality coming from the piped -> I use might not because I am not 100% sure. But you can support this argument by citing a paper which found the same thing. For example, you can cite this paper that I suggest above: https://www.nature.com/articles/s41545-018-0012-z

-> study -> this study found that …

**Conclusion**

**Important! :**

1. All paragraphs in this chapter are actually discussion. So move them up and please re-structure or re-write them all.
2. Almost all sentences are repetition of the previous chapter! Especially line 314 – 330.
3. Conclusion is not strong! Conclusion should give a brief summary of all the chapter from introduction to discussion and not just recommendation. And the last paragraph does not show a good conclusion.

-306 -> add citation

– 308 -> this is repetition of the introduction! delete

– 310 -> confusing sentence! Re-write!

-> delete "comprehensive". I don't know the reason why you say that this study is a comprehensive study

-> better innitiative -> what do you mean by better innitiative? So the government innitiative is wrong or worse before? Find other words!

-> to use safe drinking water source -> which source are safe? study also found that improved sources (like piped or protected well) are not always 100% safe. Check this: doi: 10.2471/BLT.13.119594

– 323 -> do you mention all predictors variable?

– 332 -> which results support your argument?

– 334 -> confusing sentence!

– 336 -> this is an interesting finding. But you need to ellaborate and support more with literature.

---

## Author Comment (AC1) · 3 Apr 2020

Thank you indeed for the valuable comments and based on these I would revise my paper. I would like to answer two very important issues raised:

It seems that author treat predictor variable as continuous in multinominal logit but then categorical in logit and this means that the analysis is wrong. Because if the predictors are categorical, table 2 should look like table 3 (all levels in the predictor variables have their own results or they are dummy variables). Please correct me if I am wrong. if I am correct but please re-do the analysis or update the table if I am correct.

Reply: Basically, in table 2 we have six categories of water sources (multiple categories) so the multinomial model has been adopted. A household will use water from

one of these six categories. Here first catgory i.e. water from filtration is a base category and we are comparing the coefficient (Relative risk ratios)of other variables with them. Wherein in table 3 we had only a dummy variable that either household is using water treatment method or not. So here the logit model is adopted.

5. The statistical analysis looks doubtful. Usually researcher use p value < 0.05, but author also consider p value < 0.1 as significant. Please give your reason for this in the methods section.

Reply: In the analysis, I had used the p-value of 0.05 only because if you see both the table there is only 3 occians where variable are significant at 0.01 level It would not affect the results to a great extent. So Iin the revised draft I would restrict the criteria to only P<0.05.

---

## Referee Comment (RC2) · Anonymous Referee #2 · 24 Apr 2020

Summary: using PDHS survey for 2017-18 the author find significant influence of some household characteristics on the choice of household water treatment. The author then goes on to emphasize the important role of awarenes campaigns in nudging households to treat their water at home.

Comments: the author should carefully edit the paper and present the tables in a better manner before resubmission. Poor English at places made it difficult to understand certain arguments such as reagrding the estimated effects of household size on treating household water. The research design in itself is not as innovative, since such studies have been conducted many times before in different contexts. The use of multinomial logit and logit model was not well motivated and its use was rather not-so-innovative as well. For example, why not probit model? Perhaps the author can bring in the as-

sumptions that he is making when using logit models. Results obtained are also as expected. However his use of the independent variables was well motivated by literature and challenges that Pakistani society faces in terms of access of clean drinking water. Also, use of multiple household characteristics in such a manner and its impact on use of household water treatment is perhaps new and of interest to development authorities and policy makers in Pakistan. The author should therefore make serious effort on discussing its implications for actions that policy makers can take than just focusing on awareness campaign.

---

## Author Comment (AC2) · 26 Apr 2020

Thank you indeed for the valuable comments. I will try to incorporate them in the revised draft.

---

## Author Response (AR1)

| Topical Editor | Author Responses |
|---|---|
| The literature study should not be a summary of the work of the various, but should be incorporated in the introduction, to better define the knowledge gap and the objective of the study | As desired introduction and literature review has been merged |
| The results and discussion chapter should contain a reflection on literature. What is new, what is the same as others found etc. | As desired findings have been linked with past studies on the subject. |
| Peer Review # 1 | Author Responses |
| **Major Comments** | |
| Please make the introduction and literature concise. Now you have 3+ pages of it. Please make it maximum 2 pages. That's possible. Delete unnecessary information. Make it concise. | In the light of comments introduction and literature review has been merged and shortened. |
| I think the citation and reference's style are not well reported. Please edit it following the journal's standard. Check the example in the website: https://www.drinking-water-engineering-and-science.net/for_authors/manuscript_preparation.html | Citation and references has been updated according to the journal format. |
| The conclusion is not strong enough! And the conclusion chapter is not really conclusion, but there are many repetitions from the discussion chapter. | Conclusion has been updated and important policy implications along with future research directions are added. |
| **Minor Comments** | Numerous changes in each line and page were suggested. I incorporated all these changes. |
| I think that author needs to re-write this abstract. Please consider my comments for other chapters when re-writing the abstract. | Abstract is updated in line with the comments |
| Change water purification and other related terms to HWT. Change purifying water to treated water. | Needful has been done. |
| You can add this paper as an extra citation: https://www.nature.com/articles/s41545-018-0012-z | Paper is cited. |

| | |
|---|---|
| **Summary of chapter 2** : authors can make this section more concise. Don't need to mention all significant variables that are found in those studies, including negative or positive correlation. You can discuss that when you relate your findings and their findings. But don't need to be detail in this chapter. See for example the paper from Daniel et al. (2019) (one of the papers that you cite) how they wrote all the factors very briefly in section 2.2 only in 1 paragraph. | In the light of the comments literature review is updated and made part of introduction |
| Make clear distinction between dependent and independent variable. For example, you can make different sub-title for them. | Dependent and independent variables have been distinguished. |
| If you write your assumption or hypothesis in several variables (age of household head and level of education), you need to make it in all variables, e.g., you don't make it for wealth of household. | Hypothesis have been added in all the variables. |
| Be careful of using present and past tense! Please check it. change the numbering to 3.1, 3.2, … | Needful done. |
| I would say that your classification is not scientifically acceptable. For example, why don't you use the old definition of JMP, for example: improved, unimproved, surface water? But I don't want you to re-do all your analysis. So, suggest to change the name for these 6 classifications. For example, no 1: just call it "bottled water", no 3: well, no 4: unprotected well, no 5: surface water, no 6: bough water from commercial entities | In the light of the comments the classifications have been changed as per following details:

1bottled/filtered water" 2. Piped water 3 protected well, 4 unprotected well 5: surface water, 6: water bough from commercial entities |
| The discussion is not deep enough. Author only describe the findings one-by-one for each predictor variable and don't relate the findings to bigger context or other studies. Even there is no discussion which predictor is the most important one. | As desired findings have been linked with past studies on the subject. |
| Sometimes author write unnecessary words result in long sentence. Please consider to write it more concise. | Needful has been done. |
| It seems that author treat predictor variable as continuous in multinominal logit but then categorical in logit and this means that | Basically, in table 2 we have six categories of water sources (multiple categories) so the |

| | |
|---|---|
| the analysis is wrong. Because if the predictors are categorical, table 2 should look like table 3 (all levels in the predictor variables have their own results or they are dummy variables). Please correct me if I am wrong. if I am correct but please re-do the analysis or update the table if I am correct. | multinomial model has been adopted. A household will use water from one of these six categories. Here first category i.e. bottled/filtered water is a base category and we are comparing the coefficient (Relative risk ratios)of other variables with them. Wherein in table 3 we had only a dummy variable that either household is using water treatment method or not. So here the logit model is adopted. |
| The statistical analysis looks doubtful. Usually researcher use p value < 0.05, but author also consider p value < 0.1 as significant. Please give your reason for this in the methods section. | As desired in the revised draft only P value< 0.05 is used |

| Peer Review # 2 | Author Responses |
|---|---|
| The author should carefully edit the paper and present the tables in a better manner before resubmission. | Tables are revisited and updated |
| Poor English at places made it difficult to understand certain arguments such as regarding the estimated effects of household size on treating household water. | In the light of the comments proofreading is done and tried level best to make the paper understandable. |
| The research design in itself is not as innovative, since such studies have been conducted many times before in different contexts. The use of multinomial logit and logit model was not well motivated and its use was rather not-so-innovative as well. For example, why not probit model? Perhaps the author can bring in the assumptions that he is making when using logit models. Results obtained are also as expected. However, his use of the independent variables was well motivated by literature and challenges that Pakistani society faces in terms of access of clean drinking water. Also, use of multiple household characteristics in such a manner and its impact on use of household water treatment is perhaps new and of interest to development  authorities and policy makers in Pakistan. **The author should therefore make serious effort on discussing its implications for actions that policy makers can take than just focusing on** | In the light of the comments conclusion section has been updated and important policy implications along with future research directions are added. |

[revised manuscript text omitted]